# Effect of Breastfeeding and Its Duration on Impaired Fasting Glucose and Diabetes in Perimenopausal and Postmenopausal Women: Korea National Health and Nutrition Examination Survey (KNHANES) 2010–2019

**DOI:** 10.3390/medicines8110071

**Published:** 2021-11-12

**Authors:** Byung-Soo Kwan, In-Ae Cho, Ji-Eun Park

**Affiliations:** 1Department of Internal Medicine, Division of Gastroenterology and Hepatology, Samsung Changwon Hospital, Sungkyunkwan University School of Medicine, Changwon-si 51353, Korea; Byungsoo2459.kwan@samsung.com; 2Department of Obstetrics and Gynecology, Gyeongsang National University Hospital, Jinju-si 52727, Korea; iacho@gnuh.co.kr; 3Department of Obstetrics and Gynecology, Gyeongsang National University Changwon Hospital, Changwon-si 51472, Korea; 4College of Medicine, Gyeongsang National University, Jinju-si 52828, Korea

**Keywords:** breastfeeding, impaired fasting glucose, diabetes, perimenopause, postmenopause

## Abstract

Background and Objectives: To examine the effect of maternal breastfeeding on the subsequent risk of diabetes in parous Korean women aged >50 years. Materials and Methods: A total of 14,433 participants from the Korea National Health and Nutrition Examination Survey (KNHANES) were included. The subjects were divided into three groups: normal, impaired fasting glucose, and diabetes. The adjusted odds ratios (ORs) for impaired fasting glucose (IFG) and diabetes were assessed using multivariate logistic regression. Results: A total of 2301 (15.94%) women were classified as having diabetes, and 3670 (25.43%) women were classified as having impaired fasting glucose. Breastfeeding was associated with an OR for diabetes of 0.76 (95% confidence interval (CI): 0.61, 0.95) compared with non-breastfeeding after adjustment for possible confounders in the multivariable logistic regression analysis. Breastfeeding for 13–24 months was associated with an OR of 0.68 (95% CI, 0.5, 0.91), and breastfeeding for 25–36 months was associated with an OR of 0.68 (95% CI, 0.52, 0.87) for diabetes compared with breastfeeding for <1 month in the multivariable logistic regression analysis. Conclusions: Our results suggest that long-term breastfeeding, particularly breastfeeding for 13–36 months, may be associated with a lower risk for diabetes later in life.

## 1. Introduction

Diabetes is a major global public health issue that increases the prevalence of metabolic diseases that cause microvascular and macrovascular diseases [1]. The increasing burden of diabetes is a major healthcare concern worldwide. According to the Global Burden of Disease (GBD) Study 2017, approximately 462 million individuals are affected by diabetes, accounting for 6.28% of the world’s population. The incidence peaks at approximately 55 years of age [2]. The global prevalence of diabetes is projected to increase to 552 million by 2030 [3]. Public health and clinical preventive measures are necessary. Therefore, it is important to understand the risk factors for diabetes and make efforts to lower the risk.

Many previous studies have reported on the benefits of breastfeeding in babies, including reductions in infectious diseases, sudden infant death syndrome, allergic diseases, obesity, hypertension, and neurodevelopmental disorders [4,5]. Additionally, multiple previous studies have shown that breastfeeding in mothers reduces the risks of cardiovascular disease, metabolic syndrome, and breast and ovarian cancer [6,7,8,9].

It has been suggested that breastfeeding lowers the risk of subsequent maternal diabetes through several potential mechanisms, e.g., increasing maternal energy expenditure, improving insulin sensitivity and glucose metabolism, and affecting lipid metabolism [10,11,12,13,14]. Previous epidemiological studies based on this premise have investigated the association between breastfeeding and diabetes, and the results showed that women who had never breastfed had an increased risk of diabetes [10,11].

The risk and preventive factors for diabetes in women are diverse. In particular, it has been found that the transition to menopause, which occurs just before the average age of 50 years in women, has a significant impact on the onset of diabetes [12]. However, previous studies have mostly been conducted in young or middle-aged women. To date, there have been very few studies related to breastfeeding and diabetes risk in people older than 50 years of age, which is close to the menopausal transition period. To elucidate the clinical significance of breastfeeding among mothers and to encourage actual breastfeeding, large-scale population-based studies that include other risk factors or preventive factors on the effect of breastfeeding on the prevention of diabetes are needed. Additionally, there is currently a lack of research providing definitive evidence regarding the duration of breastfeeding and its effect on the risk of diabetes in mothers. To the best of our knowledge, there has been only one study on the association between the duration of breastfeeding and maternal diabetes in postmenopausal women; that study showed that breastfeeding for more than 3 months lowered the risk of diabetes [13].

Therefore, the purpose of this study was to determine whether breastfeeding and its duration are related to the occurrence of impaired fasting glucose and diabetes in women older than 50 years using data from the Korea National Health and Nutrition Examination Survey (KNHANES) from 2010 to 2019 and to identify other factors related to the occurrence of impaired fasting glucose and diabetes.

## 2. Materials and Methods

### 2.1. Study Participants and Design

This study employed a cross-sectional design. All data is available in the KNHANES database (http://knhanes.cdc.go.kr/ Accessed 12 July 2021). The data for this study was derived from the KNHANES 2010–2019. The KNHANES has been performed periodically since 1998 to evaluate the health and nutritional status of the Korean population. Participants were selected using proportional allocation systematic sampling with multistage stratification.

A total of 80,861 persons participated in the KNHANES 2010–2019. Among them, we identified 19,156 women over the age of 50 and analyzed 15,699 women whose breastfeeding and diabetes risk could be identified. We analyzed 14,917 parous women aged over 50 years with complete data on reproductive factors and clinical variables (Figure 1). Each participant provided written informed consent.

Participants were categorized into three groups: normal, impaired fasting glucose, and diabetes. Each variable was compared.

### 2.2. Study Variables

Demographic data, health-related factors, medical history and obstetrics and gynecologic history were obtained from self-report questionnaires and personal interviews conducted by trained staff. Education level was defined as the highest level of formal education completed as of the date of the interview. This study categorized education level into four levels: elementary school or less, middle school, high school, and college or more. For income, this study used an equivalized monthly household income calculation ([monthly overall household income] [household size]^−0.5^) and divided the participants into four quartiles.

Health-related factors included cigarette smoking, alcohol intake, and aerobic or muscle exercise. Aerobic exercise was measured according to the performance of aerobic activity recommended by the WHO guidelines, and those who engaged in at least 150 min of moderate-intensity activity per week or 75 min or more of a combination of moderate-intensity and high-intensity activity per week were classified into the aerobic exercise group. Regarding muscle exercise, participants were asked, “How many days per week do you perform muscle exercises such as push-ups, sit-ups, dumbbell exercises, weightlifting, or horizontal bar lifting?” The strength training group consisted of subjects who performed resistance exercise at least twice a week.

The obstetric and gynecological history included menopause, number of pregnancies and use of oral contraceptives (OCs). Menopause was categorized as ‘yes’ or ‘no’. OC use was also categorized into two groups: OC use for ≥1 month in a lifetime or not. The duration of breastfeeding was defined as the total period of breastfeeding in a woman’s life. This information was obtained from the following open-ended question: ‘‘How long did you feed your children breast milk?’’ For women who had breastfed, a questionnaire was administered to collect information about the number of children who were breastfed and the total duration of breastfeeding. The subjects were divided into six groups according to the duration of breastfeeding: <1, <12, 13–24, 25–36, 37–48, and ≥49 months.

Trained medical staff performed anthropometric measurements following a standardized procedure. Waist circumference was measured to the nearest 0.1 cm at the narrowest point between the lower border of the rib cage and iliac crest after normal expiration. Height and body weight were measured to the nearest 0.1 cm and 0.1 kg, respectively, while participants wore light clothing without shoes. Body mass index (BMI) was calculated as weight (kg) divided by the square of height (m^2^) and categorized as underweight (<18.5 kg/m^2^), normal (≥18.5 ~ <23 kg/m^2^), overweight (≥23 ~ <25 kg/m^2^), or obese (≥25 kg/m^2^). Blood pressure was measured three times by trained nurses using a mercury sphygmomanometer (Baumanometer; Baum Co., Copiague, NY, USA); participants were in a seated position with the arm supported at heart level after 5 min of rest. All participants fasted for at least 8 h before blood sampling. Plasma glucose and total cholesterol were directly measured using a Hitachi Automatic Analyzer 7600 (Hitachi, Tokyo, Japan).

Participants were divided into three groups according to diabetes status. Diabetes was defined as a previous diagnosis or use of medication for diabetes based on self-reports or survey results, a fasting plasma glucose (FPG) level of ≥126 mg/dL (7.0 mmol/L) and/or a glycated hemoglobin (HbA1c) level of ≥6.5%. Impaired fasting glucose was defined as an FPG level of 100–125 mg/dL (5.6–6.9 mmol/L) and an HbA1c level of <6.5%. Normal was defined as an FPG level of <100 mg/dL (5.6 mmol/L). Patients were divided into three groups according to hypertensive status. Hypertension was defined as systolic blood pressure (SBP) ≥140 mmHg, diastolic blood pressure (DBP) ≥90 mmHg or the use of antihypertensive medication. Prehypertension was defined as 120 mmHg ≤ SBP < 140 mmHg or 80 mmHg ≤ DBP < 90 mmHg. Normal blood pressure was defined as SBP < 120 mmHg and DBP < 80 mmHg. Dyslipidemia was defined according to total cholesterol (TC). Desirable TC was defined as ≤200 mg/dl, borderline high was defined as 200–239 mg/dl, and high was defined as ≥240 mg/dl or the use of antidyslipidemic medication.

### 2.3. Statistical Analysis

To integrate 10 years of data, an integrated weight was applied, and one data point was produced. Comparisons of continuous variables among the three groups according to glucose status were performed with svyglm, and proportions of categorical variables were compared using the Rao-Scott chi-square test of svychisq. To analyze the risk of fasting blood glucose impairment and diabetes with reference to the normal controls, univariate analysis was performed according to each investigated variable. Multivariate logistic regression analysis was performed to evaluate the risk of impaired fasting glucose and diabetes according to breastfeeding history. Variables with a statistically significant association on univariate analysis were included in a multivariate logistic regression analysis. Among the significant variables, only one variable with a strong correlation (correlation coefficient of 0.7 or higher) was selected and included in the analysis to solve the multicollinearity problem. We considered 11 confounding variables (age, income, education, alcohol consumption, smoking, BMI, hypertension, total cholesterol, menstruation, oral contraceptive use, and exercise), for which we adjusted in the regression. In addition, the risk of impaired fasting blood glucose and diabetes according to the duration of breastfeeding was also analyzed. All analyses were conducted using R software version 4.0.3 (R Core Team, R Foundation for Statistical Computing, Vienna, Austria, 2020). For all analyses, *p*-values < 0.05 were considered statistically significant.

### 2.4. Ethics

Access to the KNHANES data was acquired after receiving approval by the Institutional Review Board (IRB) of the Korea Center for Disease Control and Prevention (CDC). This study is a retrospective study that used and analyzed the data from the KNHANES survey; therefore, approval from the IRB was not required. Because the dataset did not include personal information and participants gave consent to the KNHANES, further ethical approval for the use of open KNHANES data was exempted from the IRB.

## 3. Results

The mean age of the participants (N = 14,433) was 63.4 ± 8.8 years; 13,027 (90.26%) had ever breastfed, and the mean duration of breastfeeding was 40.3 ± 39.9 months. In total, 2301 (15.94%) participants had diabetes, and 3670 (25.43%) participants had impaired fasting glucose. As shown in Table 1, statistically significant differences were observed among the normal, impaired fasting glucose and diabetes groups for age, socioeconomic status (house income, educational level), lifestyle factors (alcohol consumption, exercise), body measurements (waist circumference, body mass index (BMI)), hypertensive status, total cholesterol level, number of pregnancies, age at last birth, menopause, history of OC use, breastfeeding experience, number of children who were breastfed, and duration of breastfeeding. The group with diabetes had more pregnancies and a longer breastfeeding duration than the normal group.

The univariate analysis results regarding the risk of impaired fasting glucose and diabetes compared to normal controls are shown in Table 2. Factors significantly related to the risk of diabetes included older age, low socioeconomic status (house income, educational level), high waist circumference or BMI, prehypertension or hypertension, high total cholesterol (>240) or the use of antidyslipidemic medication, menopause, a larger number of pregnancies, a history of OC use, a history of breastfeeding, having three or more children, and breastfeeding for more than 37 months. On the other hand, a low risk of diabetes was associated with consuming alcohol, a total cholesterol level of 200 or higher, a total breastfeeding period of 13–24 months, and a sedentary lifestyle.

In the multivariate logistic regression analysis, after adjustment for variables that were significant in the univariate analysis, breastfeeding was inversely associated with the risk of maternal diabetes (OR 0.76; 95% CI, 0.61, 0.95; *p* = 0.016) (Table 3); however, there was no significant association with maternal IFG (OR 0.88; 95% confidence interval: 0.76, 1.04; *p* = 0.154). Other risk factors related to diabetes were age (OR, 1.03; CI, 1.02, 1.04; *p* < 0.001), a lower income level (in the lower half) (OR, 1.26; CI, 1.04, 1.53; *p* = 0.017), smoking (OR, 1.28; CI, 1.02, 1.61; *p* < 0.032), a higher BMI (OR, 1.14; CI, 1.12, 1.16; *p* < 0.001), hypertension (OR, 2.78; CI, 2.34, 3.3; *p* < 0.001), high TC ≥240 mg/dl or the use of antidyslipidemic medication (OR, 1.47; CI, 1.28, 1.68; *p* < 0.001), menopause (OR, 1.62; CI, 1.13, 2.31; *p* = 0.008), and no exercise (OR, 1.36; CI, 1.04, 1.78; *p* = 0.024). On the other hand, alcohol consumption (OR, 0.80; CI, 0.71, 0.92; *p* = 0.001) and borderline high TC (200–239 mg/dl) (OR, 0.55; CI, 0.46, 0.64; *p* < 0.001) were associated with a lower risk of maternal diabetes.

Table 4 shows the results of the multivariate logistic regression analysis of the maternal risk of impaired fasting blood glucose and diabetes according to the duration of breastfeeding, with adjustment for significant variables. It was identified that the risk of diabetes was decreased when breastfeeding was carried out for a period of 13 to 36 months. In the subanalysis, an OR of 0.68 (95% CI: 0.5, 0.91; *p* = 0.011) for diabetes in the 13–24 month breastfeeding group and an OR of 0.67 (95% CI: 0.52, 0.87; *p* = 0.002) for diabetes in the 25–36 month breastfeeding group were found.

## 4. Discussion

In this nationwide study, we found that mothers who had breastfed had a significantly lower risk of diabetes later in the peri- and postmenopausal periods than women who had not breastfed, independent of other diabetes risk factors. To the best of our knowledge, this study is the first to determine the association between the risk of diabetes and breastfeeding and breastfeeding duration in transitional or postmenopausal women over 50 years of age. Among the participants, it was confirmed that a breastfeeding period between 13 and 36 months was associated with a reduction in the risk of diabetes. In addition, age, low socioeconomic status, current smoking, obesity, elevated blood pressure, high TC (≥240 mg/dl) or the use of antidyslipidemic medication, menopause, and no exercise were associated with an increased risk of diabetes. Interestingly, alcohol consumption and a borderline high TC level (200–239 mg/dl) were associated with a lower risk of diabetes.

The Korean Diabetes Association (KDA) has published Diabetes Fact Sheets (DFSs) since 2012 based on the KNHANES. According to the 2020 announcement [14], 11.8% of Korean women over 30 years of age had diabetes, and the prevalence rate among adults 65 years and older was 27.5%. The prevalence of IFG was 22% in women over 30 years of age and 28.1% in women over 65 years of age. This report, developed from the same source as that used in our study, shows that the risks of diabetes and IFG increase with age. The prevalence of diabetes among those over 50 years old in Korea was not significantly different from the prevalence of diabetes in those over 50 years old in Mexico, which was reported to be 19.34% in 2017 [15].

The association between breastfeeding and diabetes has been identified in previous studies. Stuebe et al. [16] found that a longer duration of breastfeeding decreased the risk of diabetes in two large (N = 121,700, N = 116,671) cohorts of young and middle-aged women (aged 30–55 and 25–42 years, respectively) in the US. Villegas et al. [10] found that among 62,095 middle-aged women (aged 40 years and older) in Shanghai, China, women who breastfed tended to have a lower risk of diabetes than women who never breastfed (RR = 0.88; 95% CI, 0.76–1.02; *p* = 0.08). Zhang et al. [11], in a cohort of 9128 middle-aged women (aged 40 years and older) in Beijing, China, reported that women who had not breastfed had a higher risk of diabetes than women who had breastfed (OR = 1.30; 95% CI 1.11–1.53). Our study also confirmed the association between breastfeeding and maternal diabetes. However, our study population was different from those in previous studies, as it comprised women over the age of 50 years.

In addition, our study confirmed that a breastfeeding duration of 13–36 months was associated with a lower risk of maternal diabetes than a history of breastfeeding less than 1 month. Zhang et al. [11] reported a lower risk of maternal diabetes in the group that breastfed for more than 0 and up to 6 months and in the group that breastfed for 6–12 months than in the non-breastfeeding group, and Luo et al. [13] reported that breastfeeding for more than 24 months lowered the risk of diabetes. In our study, there were no clear effects on the risk of diabetes in the groups lactating for less than 12 months and more than 37 months. The lack of an association of long-term breastfeeding for more than 37 months with diabetes was probably due to increased parity, which contributed to the long lactation period. Pregnancy is a condition in which the mother’s fasting and postprandial blood sugar fluctuates, insulin secretion increases, and insulin resistance worsens [17]. This suggests that an increase in parity contributes to an increased risk of diabetes.

The mechanisms underlying the link between breastfeeding and diabetes are still unclear. Several probable mechanisms reported in the literature may explain the inverse relationship between breastfeeding and the risk of type 2 diabetes. First, breastfeeding is effective for weight loss and preventing obesity by increasing energy consumption [18], which can help lower the risk of diabetes. Additionally, breastfeeding exerts a positive effect on glucose metabolism by lowering fasting blood sugar and improving insulin resistance in both human and animal studies [19,20]. There are also reports showing that postpartum lactation is associated with increased pancreatic beta-cell function in women with gestational diabetes [21]. Breastfeeding has also been linked to improved lipid metabolism [22] and increased maternal adipokine levels, including peptide YY and ghrelin [23], which may reduce the risk of diabetes and metabolic disorders. These previous reports further support our findings.

The WHO and the United Nations International Children’s Emergency Fund (UNICEF) recommend that children be breastfed within one hour after birth and be exclusively breastfed for the first 6 months of life. From the age of 6 months, children should begin eating safe and adequate complementary foods while continuing to breastfeed for 2 years or more [24]. Nevertheless, due to various social and personal reasons, the rate of breastfeeding for more than 6 months in Korea was 11.4% in 2012 [25]. Our findings of an association between breastfeeding for more than 12 months and a decreased risk of diabetes may serve as a basis for educating and persuading mothers to breastfeed more actively and for a longer period of time.

The strength of this study compared to the previous population-based studies in the US and China mentioned above is that in addition to breastfeeding, we examined whether several previously identified factors related to the occurrence of diabetes were actually related. In our study, in addition to breastfeeding-related factors, age, socioeconomic status, smoking status, obesity, exercise, and high blood pressure were also associated with the risk of developing diabetes. This is in line with previous studies [26,27] and suggests that it is possible to prevent the occurrence of diabetes.

Previous clinical studies have reported that increased cholesterol levels decrease glucose tolerance and that high TC to high-density lipoprotein cholesterol (HDL-C) ratios increase the risk of type 2 diabetes [28]. In contrast, recent studies have also reported that patients with familial hypercholesterolemia with high levels of low-density lipoprotein cholesterol (LDL-C) have a lower risk of diabetes [29]. In this study, a low OR for the association between borderline high total cholesterol and diabetes was identified.

This study has some limitations. First, the cross-sectional study design does not allow the establishment of a causal relationship between breastfeeding and diabetes. Second, this study may be subject to recall bias because data were collected based on the memories of individuals’ breastfeeding periods and exercise patterns. Third, although type 1 diabetes and type 2 diabetes have different etiologies for progressive pancreatic beta-cell failure, we could not determine which type of diabetes any given subject had in this study. Fourth, the results of tests such as the oral glucose tolerance test (OGTT), which is used to assess beta-cell function and insulin resistance [30], could not be determined. Finally, due to the lack of data, we could not check for a family history of diabetes or a history of gestational diabetes, both of which are closely related to the development of diabetes. Despite these limitations, our study’s strength is that it analyzed 10 years of data from a large national representative sample, identified multiple relevant covariate factors related to diabetes, and adjusted for confounding factors. We are confident that these results will help establish evidence-based health policies that promote women’s health through breastfeeding.

## 5. Conclusions

A nationally representative, population-based cross-sectional study found that peri- and postmenopausal women who breastfed for a long period of time had a lower risk of diabetes than women who had not breastfed. The risk of diabetes was lower in women who had breastfed for 13 to 36 months. The relationships of breastfeeding for <12 months and >37 months with diabetes risk were not statistically significant. Our results suggest that long-term breastfeeding for more than 1 year may lower the risk of diabetes in mothers, and this can be a strong basis for establishing women’s health policies by encouraging mothers to breastfeed.

## Figures and Tables

**Figure 1 medicines-08-00071-f001:**
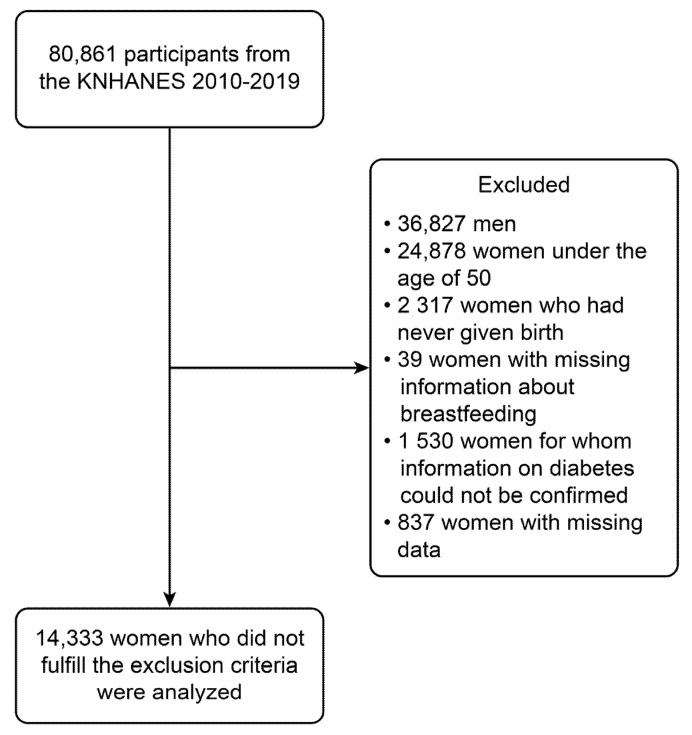
Participant flow diagram for final analysis.

**Table 1 medicines-08-00071-t001:** Characteristics of the study participants by glucose status.

	Total (N = 14,433)	Normal (N = 8462)	Impaired Fasting Glucose (N = 3670)	Diabetes (N = 2301)	*p*-Value
Age (years), mean (SD)	63.4 ± 8.8	**62.2 ± 8.7**	**63.7 ± 8.7**	**67.1 ± 8.3**	**<0.001**
Income, N (%)					**<0.001**
Q4	3235	**2135 (66%)**	**786 (24.3%)**	**314 (9.7%)**	
Q3	3103	**1903 (61.3%)**	**782 (25.2%)**	**418 (13.5%)**	
Q2	3762	**2135 (56.8%)**	**1003 (26.7%)**	**624 (16.6%)**	
Q1	4333	**2289 (52.8%)**	**1099 (25.4%)**	**945 (21.8%)**	
Education, N (%)					**<0.001**
College or higher	1500	**1043 (69.5%)**	**348 (23.2%)**	**109 (7.3%)**	
High school	3445	**2199 (63.8%)**	**880 (25.5%)**	**366 (10.6%)**	
Middle school	2413	**1434 (59.4%)**	**624 (25.9%)**	**355 (14.7%)**	
Less than elementary school	7075	**3786 (53.5%)**	**1818 (25.7%)**	**1471 (20.8%)**	
Alcohol consumption, N (%)					**<0.001**
No alcohol consumption	4021	**2245 (55.8%)**	**960 (23.9%)**	**816 (20.3%)**	
Alcohol consumption	10,412	**6217 (59.7%)**	**2710 (26%)**	**1485 (14.3%)**	
Smoking, N (%)					0.053
Nonsmoking	13,533	7956 (58.8%)	3459 (25.6%)	2118 (15.7%)	
Current smoking	900	506 (56.2%)	211 (23.4%)	183 (20.3%)	
Waist circumference	82.2 ± 9.1	**80.3 ± 8.5**	**84.0 ± 9.0**	**86.6 ± 9.3**	**<0.001**
BMI	24.3 ± 3.3	**23.7 ± 3.0**	**24.9 ± 3.3**	**25.4 ± 3.6**	**<0.001**
BMI (categorical)					**<0.001**
BMI < 18.5	291	**217 (74.6%)**	**50 (17.2%)**	**24 (8.2%)**	
18.5 ≤ BMI < 25.0	8816	**5761 (65.3%)**	**1943 (22%)**	**1112 (12.6%)**	
25.0 ≤ BMI < 30.0	4593	**2238 (48.7%)**	**1424 (31%)**	**931 (20.3%)**	
30.0 ≤ BMI	733	**246 (33.6%)**	**253 (34.5%)**	**234 (31.9%)**	
Hypertension					**<0.001**
Normal	4118	**3021 (73.4%)**	**810 (19.7%)**	**287 (7%)**	
Prehypertension	3362	**2146 (63.8%)**	**835 (24.8%)**	**381 (11.3%)**	
Hypertension	6953	**3295 (47.4%)**	**2025 (29.1%)**	**1633 (23.5%)**	
Total cholesterol	199.3 ± 38.4	**202.0 ± 36.1**	**203.5 ± 39.0**	**182.5 ± 41.1**	**<0.001**
Total cholesterol (categorical)					**<0.001**
<200	5113	**3149 (61.6%)**	**1127 (22%)**	**837 (16.4%)**	
200≤, <240	4461	**2895 (64.9%)**	**1183 (26.5%)**	**383 (8.6%)**	
>240	4859	**2418 (49.8%)**	**1360 (28%)**	**1081 (22.2%)**	
Menstruation					**<0.001**
Menstruation	975	**702 (72%)**	**222 (22.8%)**	**51 (5.2%)**	
Menopause	13,458	**7760 (57.7%)**	**3448 (25.6%)**	**2250 (16.7%)**	
Number of pregnancies	4.5 ± 2.2	**4.4 ± 2.1**	**4.5 ± 2.1**	**5.1 ± 2.4**	**<0.001**
Breast feeding, N (%)					**0.002**
No	1406	**881 (62.7%)**	**361 (25.7%)**	**164 (11.7%)**	
Yes	13,027	**7581 (58.2%)**	**3309 (25.4%)**	**2137 (16.4%)**	
Breastfeeding duration (months), mean (SD)	40.3 ± 39.9	**37.5 ± 39.1**	**39.8 ± 38.3**	**51.4 ± 43.6**	**<0.001**
Breastfeeding duration (months) categorical, N (%)					**<0.001**
Less 1 month	1409	**883 (62.7%)**	**361 (25.6%)**	**165 (11.7%)**	
Less than 12 months	1120	**748 (66.8%)**	**263 (23.5%)**	**109 (9.7%)**	
13–⁠24 months	1297	**839 (64.7%)**	**319 (24.6%)**	**139 (10.7%)**	
25–⁠36 months	3257	**2046 (62.8%)**	**826 (25.4%)**	**385 (11.8%)**	
37–⁠48 months	2240	**1270 (56.7%)**	**598 (26.7%)**	**372 (16.6%)**	
More than 49 months	5110	**2676 (52.4%)**	**1303 (25.5%)**	**1131 (22.1%)**	
Number of children breastfed, mean (SD)	2.5 ± 1.6	**2.4 ± 1.5**	**2.6 ± 1.6**	**2.9 ± 1.7**	**<0.001**
Number of children breastfed, N (%)					**<0.001**
0	1406	**881 (62.7%)**	**361 (25.7%)**	**164 (11.7%)**	
1 or 2	6609	**4142 (62.7%)**	**1636 (24.8%)**	**831 (12.6%)**	
3 or 4	4826	**2597 (53.8%)**	**1286 (26.6%)**	**943 (19.5%)**	
More than 5	1592	**842 (52.9%)**	**387 (24.3%)**	**363 (22.8%)**	
Oral contraceptive use					**<0.001**
No	11,229	**6696 (59.6%)**	**2846 (25.3%)**	**1687 (15%)**	
Yes	3204	**1766 (55.1%)**	**824 (25.7%)**	**614 (19.2%)**	
Age at last birth	29.6 ± 4.4	**29.5 ± 4.3**	**29.6 ± 4.4**	**29.8 ± 4.7**	**0.007**
Exercise, N (%)					**<0.001**
Muscle and aerobic exercise	1036	**687 (66.3%)**	**247 (23.8%)**	**102 (9.8%)**	
Only aerobic exercise	4607	**2768 (60.1%)**	**1145 (24.9%)**	**694 (15.1%)**	
Only muscle exercise	818	**513 (62.7%)**	**209 (25.6%)**	**96 (11.7%)**	
No exercise	7972	**4494 (56.4%)**	**2069 (26%)**	**1409 (17.7%)**	

Note: Values are presented medians ± ranges or n (%); BMI, body mass index; N, number of participants; %, percentage. The bold values are values with significant differences (*p* < 0.05).

**Table 2 medicines-08-00071-t002:** Univariate logistic regression analysis results of risks for impaired fasting glucose and diabetes.

	Risk for Impaired Fasting Glucose Compared to Normal			Risk for Diabetes Compared to Normal		
	OR	95% CI	*p*-Value	OR	95% CI	*p*-Value
Age	**1.02**	**(1.02, 1.03)**	**<0.001**	**1.07**	**(1.06, 1.07)**	**<0.001**
Income						
Q4	1.00	Reference		1.00	Reference	
Q3	1.07	(0.94, 1.23)	0.312	**1.49**	**(1.23, 1.8)**	**<0.001**
Q2	**1.22**	**(1.07, 1.39)**	**0.003**	**1.96**	**(1.65, 2.33)**	**<0.001**
Q1	**1.29**	**(1.14, 1.47)**	**<0.001**	**2.99**	**(2.54, 3.52)**	**<0.001**
Education						
College or higher	1.00	Reference		1.00	Reference	
High school	1.14	(0.96, 1.34)	0.138	**1.49**	**(1.16, 1.9)**	**0.002**
Middle school	**1.24**	**(1.04, 1.48)**	**0.019**	**2.17**	**(1.67, 2.81)**	**<0.001**
Elementary school or lower	**1.45**	**(1.24, 1.68)**	**<0.001**	**3.69**	**(2.95, 4.62)**	**<0.001**
Alcohol consumption						
No alcohol consumption	1.00	Reference		1.00	Reference	
Alcohol consumption	1.01	(0.91, 1.13)	0.819	**0.62**	**(0.55, 0.69)**	**<0.001**
Smoking						
Nonsmoking	1.00	Reference		1.00	Reference	
Smoking	0.90	(0.74, 1.1)	0.312	1.22	(0.99, 1.51)	0.063
Waist circumference	**1.06**	**(1.05, 1.06)**	**<0.001**	**1.09**	**(1.08, 1.09)**	**<0.001**
BMI	**1.15**	**(1.13, 1.17)**	**<0.001**	**1.18**	**(1.16, 1.2)**	**<0.001**
BMI (categorical)						
BMI < 18.5	1.00	Reference		1.00	Reference	
18.5 ≤ BMI < 25.0	**1.44**	**(1.01, 2.07)**	**0.044**	**1.76**	**(1.06, 2.92)**	**0.028**
25.0 ≤ BMI < 30.0	**2.84**	**(1.97, 4.11)**	**<0.001**	**3.97**	**(2.4, 6.56)**	**<0.001**
30.0 ≤ BMI	**5.11**	**(3.43, 7.63)**	**<0.001**	**9.54**	**(5.58, 16.3)**	**<0.001**
Hypertension						
Normal	1.00	Reference		1.00	Reference	
Prehypertension	**1.46**	**(1.28, 1.67)**	**<0.001**	**1.87**	**(1.53, 2.28)**	**<0.001**
Hypertension	**2.36**	**(2.11, 2.64)**	**<0.001**	**5.15**	**(4.4, 6.03)**	**<0.001**
Total cholesterol	**1.00**	**(1, 1)**	**0.027**	**0.99**	**(0.98, 0.99)**	**<0.001**
Total cholesterol (categorical)						
<200	1.00	Reference		1	Reference	
200≤, <240	1.12	(1, 1.25)	0.057	**0.50**	**(0.43, 0.59)**	**<0.001**
240≤	**1.63**	**(1.46, 1.83)**	**<0.001**	**1.71**	**(1.52, 1.94)**	**<0.001**
Menstruation						
Menstruation	1.00	Reference		1.00	Reference	
Menopause	**1.37**	**(1.14, 1.64)**	**<0.001**	**3.36**	**(2.42, 4.66)**	**<0.001**
Number of pregnancies	**1.03**	**(1.01, 1.05)**	**0.006**	**1.14**	**(1.11, 1.17)**	**<0.001**
Breastfeeding, N (%)						
No	1.00	Reference		1.00	Reference	
Yes	1.10	(0.95, 1.28)	0.191	**1.41**	**(1.16, 1.7)**	**<0.001**
Breastfeeding duration (months)	**1.00**	**(1, 1)**	**<0.001**	**1.01**	**(1.01, 1.01)**	**<0.001**
Breastfeeding duration (months) categorical						
Less than 1 month	1.00	Reference		1.00	Reference	
Less than 12 months	0.87	(0.71, 1.07)	0.176	0.79	(0.59, 1.06)	0.120
13–24 months	0.93	(0.76, 1.14)	0.485	0.77	(0.59, 1.01)	0.064
25–36 months	1.04	(0.88, 1.23)	0.662	0.91	(0.73, 1.14)	0.417
37–48 months	1.13	(0.94, 1.35)	0.185	**1.47**	**(1.18, 1.83)**	**<0.001**
More than 49 months	**1.30**	**(1.11, 1.52)**	**0.001**	**2.27**	**(1.87, 2.77)**	**<0.001**
Number of children breastfed	**1.07**	**(1.04, 1.1)**	**<0.001**	**1.23**	**(1.19, 1.27)**	**<0.001**
Number of children breastfed						
0	1.00	Reference		1.00	Reference	
1 or 2	0.98	(0.85, 1.14)	0.835	0.99	(0.81, 1.21)	0.900
3 or 4	**1.28**	**(1.09, 1.51)**	**0.003**	**1.91**	**(1.56, 2.33)**	**<0.001**
More than 5	**1.24**	**(1.02, 1.51)**	**0.031**	**2.36**	**(1.88, 2.97)**	**<0.001**
Oral contraceptive use						
No	1.00	Reference		1.00	Reference	
Yes	1.08	(0.96, 1.2)	0.187	**1.33**	**(1.17, 1.51)**	**<0.001**
Age at last birth	1.01	(1, 1.02)	0.112	**1.02**	**(1.01, 1.03)**	**0.003**
Exercise						
Muscle and aerobic exercise	1.00	Reference		1.00	Reference	
Only aerobic exercise	1.16	(0.97, 1.39)	0.114	**1.73**	**(1.33, 2.26)**	**<0.001**
Only muscle exercise	1.10	(0.86, 1.42)	0.443	1.21	(0.85, 1.73)	0.293
No exercise	**1.29**	**(1.07, 1.55)**	**0.008**	**2.20**	**(1.71, 2.83)**	**<0.001**

The bold values are values with significant differences (*p* < 0.05).

**Table 3 medicines-08-00071-t003:** Multivariate logistic regression analysis results of risk for impaired fasting glucose and diabetes according to breastfeeding experience.

	Risk for Impaired Fasting Glucose Compared to Normal			Risk for Diabetes Compared to Normal		
	OR	95% CI	*p*-Value	OR	95% CI	*p*-Value
Breastfeeding						
No	1.00	Reference		1.00	Reference	
Yes	0.88	(0.76, 1.04)	0.154	**0.76**	**(0.61, 0.95)**	**0.016**
Age	**1.02**	**(1.01, 1.02)**	**<0.001**	**1.03**	**(1.02, 1.04)**	**<0.001**
Income						
Q4	1.00	Reference		1.00	Reference	
Q3	0.95	(0.82, 1.1)	0.493	1.17	(0.95, 1.43)	0.131
Q2	1.03	(0.89, 1.18)	0.709	**1.26**	**(1.04, 1.53)**	**0.017**
Q1	0.94	(0.8, 1.09)	0.399	**1.29**	**(1.06, 1.56)**	**0.011**
Education						
College or higher	1.00	Reference		1.00	Reference	
High school	1.04	(0.87, 1.24)	0.635	1.22	(0.95, 1.57)	0.119
Middle school	0.99	(0.82, 1.2)	0.919	1.28	(0.97, 1.69)	0.075
Less than elementary school	0.94	(0.78, 1.13)	0.494	1.25	(0.97, 1.61)	0.084
Alcohol consumption						
No alcohol consumption	1.00	Reference		1.00	Reference	
Alcohol consumption	**1.15**	**(1.03, 1.29)**	**0.014**	**0.80**	**(0.7, 0.91)**	**<0.001**
Smoking						
Nonsmoking	1.00	Reference		1.00	Reference	
Smoking	0.93	(0.76, 1.14)	0.510	**1.28**	**(1.02, 1.61)**	**0.032**
BMI	**1.13**	**(1.11, 1.15)**	**<0.001**	**1.14**	**(1.12, 1.16)**	**<0.001**
Hypertension						
Normal	1.00	Reference		1.00	Reference	
Prehypertension	**1.31**	**(1.15, 1.5)**	**<0.001**	**1.53**	**(1.25, 1.87)**	**<0.001**
Hypertension	**1.86**	**(1.65, 2.1)**	**<0.001**	**2.78**	**(2.34, 3.3)**	**<0.001**
Total cholesterol						
<200	1.00	Reference		1.00	Reference	
200≤, <240	1.10	(0.98, 1.23)	0.116	**0.55**	**(0.46, 0.64)**	**<0.001**
240≤	**1.40**	**(1.24, 1.57)**	**<0.001**	**1.47**	**(1.28, 1.68)**	**<0.001**
Menstruation						
Menstruation	1.00	Reference		1.00	Reference	
Menopause	1.13	(0.93, 1.37)	0.233	**1.62**	**(1.13, 2.31)**	**0.008**
Number of pregnancies	0.98	(0.95, 1)	0.099	1.03	(1, 1.06)	0.096
Oral contraceptive use						
No	1.00	Reference		1.00	Reference	
Yes	0.98	(0.87, 1.1)	0.714	1.05	(0.91, 1.2)	0.535
Age at last birth	1.01	(0.99, 1.02)	0.359	0.99	(0.98, 1.01)	0.233
Exercise						
Muscle and aerobic exercise	1.00	Reference		1.00	Reference	
Only aerobic exercise	1.08	(0.9, 1.3)	0.398	1.30	(0.98, 1.72)	0.066
Only muscle exercise	1.10	(0.85, 1.42)	0.455	0.95	(0.65, 1.39)	0.803
No exercise	1.12	(0.93, 1.36)	0.239	**1.36**	**(1.04, 1.78)**	**0.024**

The bold values are values with significant differences (*p* < 0.05).

**Table 4 medicines-08-00071-t004:** Multivariate logistic regression analysis results of risk for impaired fasting glucose and diabetes according to breastfeeding duration.

	Risk for Impaired Fasting Glucose Compared to Normal			Risk for Diabetes Compared to Normal		
	OR	95% CI	*p*-Value	OR	95% CI	*p*-Value
Breastfeeding duration						
Less than 1 month	1.00	Reference		1.00	Reference	
less than 12 months	0.87	(0.71, 1.07)	0.188	0.90	(0.65, 1.23)	0.502
13–24 months	0.85	(0.69, 1.05)	0.143	**0.68**	**(0.5, 0.91)**	**0.011**
25–36 months	0.90	(0.76, 1.08)	0.276	**0.67**	**(0.52, 0.87)**	**0.002**
37–48 months	0.89	(0.73, 1.08)	0.228	0.79	(0.61, 1.02)	0.069
More than 49 months	0.90	(0.74, 1.09)	0.269	0.84	(0.66, 1.08)	0.172
Age	**1.02**	**(1.01, 1.02)**	**<0.001**	**1.03**	**(1.02, 1.04)**	**<0.001**
Income						
Q1	1.00	Reference		1.00	Reference	
Q2	0.95	(0.82, 1.1)	0.491	1.17	(0.96, 1.43)	0.130
Q3	1.03	(0.89, 1.18)	0.711	**1.26**	**(1.04, 1.53)**	**0.018**
Q4	0.94	(0.8, 1.09)	0.401	**1.28**	**(1.05, 1.55)**	**0.013**
Education						
College or higher	1.00	Reference		1.00	Reference	
High school	1.04	(0.87, 1.24)	0.664	1.25	(0.97, 1.6)	0.088
Middle school	0.98	(0.81, 1.2)	0.857	1.31	(0.99, 1.73)	0.058
Less than elementary school	0.93	(0.77, 1.12)	0.448	1.25	(0.96, 1.62)	0.094
Alcohol consumption						
No alcohol consumption	1.00	Reference		1.00	Reference	
Alcohol consumption	**1.15**	**(1.03, 1.29)**	**0.014**	**0.80**	**(0.71, 0.92)**	**0.001**
Smoking						
Nonsmoking	1.00	Reference		1.00	Reference	
Smoking	0.94	(0.76, 1.15)	0.518	**1.29**	**(1.02, 1.61)**	**0.030**
BMI	**1.13**	**(1.11, 1.15)**	**<0.001**	**1.14**	**(1.12, 1.16)**	**<0.001**
Hypertension						
Normal	1.00	Reference		1.00	Reference	
Prehypertension	**1.31**	**(1.15, 1.5)**	**<0.001**	**1.53**	**(1.25, 1.87)**	**<0.001**
Hypertension	**1.86**	**(1.65, 2.1)**	**<0.001**	**2.77**	**(2.33, 3.29)**	**<0.001**
Total cholesterol						
<200	1.00	Reference		1.00	Reference	
200≤, <240	1.10	(0.98, 1.23)	0.114	**0.55**	**(0.46, 0.64)**	**<0.001**
240≤	**1.40**	**(1.24, 1.58)**	**<0.001**	**1.47**	**(1.29, 1.68)**	**<0.001**
Menstruation						
Menstruation	1.00	Reference		1.00	Reference	
Menopause	1.12	(0.92, 1.37)	0.241	**1.66**	**(1.16, 2.38)**	**0.006**
Number of pregnancies	0.98	(0.95, 1)	0.099	1.02	(0.99, 1.05)	0.171
Oral contraceptive use						
No	1.00	Reference		1.00	Reference	
Yes	0.98	(0.87, 1.1)	0.719	1.05	(0.91, 1.2)	0.527
Age at last birth	1.01	(0.99, 1.02)	0.361	0.99	(0.97, 1)	0.136
Exercise						
Muscle and aerobic exercise	1.00	Reference		1.00	Reference	
Only aerobic exercise	1.08	(0.9, 1.3)	0.405	1.29	(0.97, 1.71)	0.076
Only muscle exercise	1.10	(0.85, 1.42)	0.454	0.95	(0.65, 1.38)	0.776
No exercise	1.12	(0.93, 1.36)	0.239	**1.35**	**(1.03, 1.76)**	**0.028**

The bold values are values with significant differences (*p* < 0.05).

## Data Availability

Data and materials are available on reasonable request.

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
