# Peer review of "Effect of Breastfeeding and Its Duration on Impaired Fasting Glucose and Diabetes in Perimenopausal and Postmenopausal Women: Korea National Health and Nutrition Examination Survey (KNHANES) 2010–2019"

_medicines, 2021, doi:10.3390/medicines8110071_

Round 1

Reviewer 1 Report

Dear Authors,

 You will find my comments in the file attached.

Best regards.

Author Response

Reviewers' comments:

Reviewer #1

The paper entitled “Effect of Breastfeeding and Its Duration on Impaired Fasting Glucose and Diabetes in Perimenopausal and Postmenopausal Women: Korea National Health and Nutrition Examination Survey (KNHANES) 2010–2019” is interesting. However, some minor points should be considered. The manuscript requires some revision, as it contains grammatical errors. You can find several of them below.

  1. We appreciate the time and effort you have dedicated to providing insightful feedback on ways to strengthen our paper. We apologize for the extra time you have had to spend correcting our grammatical errors.
  2. Line 44: allergic diseases; and obesity - allergic diseases, obesity
  3. We appreciate these insightful comments. We have revised the text.
  • Revision: Many previous studies have reported on the benefits of breastfeeding in babies, including reductions in infectious diseases, sudden infant death syndrome, allergic diseases, obesity, hypertension and neurodevelopmental disorders [4,5].
  1. Line 66: All data are… -All data is…
  2. We appreciate your advice. We have revised the text.
  • Revision: All data is available in the KNHANES database (http://knhanes.cdc.go.kr). (Line 75)
  1. Line 67: Data for this study were… - Data for this study was…
  2. We appreciate your advice. We have revised the text.
  • Revision: The data for this study was derived from the KNHANES 2010–2019. (Line 76)
  1. Line 79-80: Participants were categorized into three groups, namely, normal, impaired fasting glucose and diabetes, and each variable was compared - Participants were categorized into three groups: normal, impaired fasting glucose and diabetes. Each variable was compared.
  2. We appreciate your advice. We have revised the text.
  • Revision: Participants were categorized into three groups: normal, impaired fasting glucose and diabetes. Each variable was compared. (Line 88-89)
  1. Line 213: between 13 and 36 months of was associated with…. - between 13 and 36 months was associated with….
  2. We appreciate your advice. We have revised the text
  • Revision: Among the participants, it was confirmed that a breastfeeding period between 13 and 36 months was associated with a reduction in the risk of diabetes. (Line 223)
  1. Line 243: in the groups who breastfed for more than 0–6 months and for more than 6– 12 months - in the groups that breastfed for more than 0–6 months and 6–12 months
  2. We appreciate your advice. We have revised the text.
  • Revision: Zhang et al. [23] reported a lower risk of maternal diabetes in the group that breastfed for more than 0 and up to 6 months and in the group that breastfed for 6–12 months than in the non-breastfeeding group, … (Line 253-255)
  1. Line 258: recommend that children start breastfeeding… - recommend that children should be breastfed…
  2. We appreciate your advice. We have revised the text;
  • Revision: The WHO and the United Nations International Children's Emergency Fund (UNICEF) recommend that children be breastfed within one hour after birth and be exclusively breastfed for the first 6 months of life. (Line 275-277)
  1. It is necessary to indicate the type of diabetes (T1D or/and T2D)
  2. Thank you for these insights. You raised an important issue, and we fully agree with you. This study used a population-based cross-sectional study; therefore, unfortunately, researchers cannot add survey items. Regarding diabetes, which is the subject of this study, only the diagnostic criteria for diabetes are provided in these raw data, and there is no history or other information that can distinguish between type 1 and type 2 diabetes; therefore, the type of diabetes cannot be specified. This point was added to the description of limitations in the discussion section.
  • Line 300-302 : Third, although type 1 diabetes and type 2 diabetes have different etiologies for progressive pancreatic beta-cell failure, we could not determine which type of diabetes any given subject had in this study.

  1. The Oral Glucose Tolerance Test (OGTT) is being used to identify the alter carbohydrate metabolism during post glucose administration and glucose utilization capacity. This test had to be carried out!
  2. We appreciate this insightful comment. You raised an important issue, and we fully agree with you. Unfortunately, however, the OGTT was not part of the population-based dataset on which this study was based. This has been added as a limitation of our study.
  • Line 302-304 : Fourth, the results of tests such as the oral glucose tolerance test (OGTT), which is used to assess beta-cell function and insulin resistance [36], could not be determined.
  1. The manuscript is based on the statistical data, and in the end of “Discussion” the author`s mentioned the limitation that complicated the study. However, the authors had to explain the relationship between diabetes and breastfeeding from the perspective of biochemistry, since there are a lot of the reasons of diabetes mellitus (e.g. stress, sedentary lifestyle, unhealthy food, cardiovascular diseases, etc.) which hadn’t been reviewed throughout the collection of statistical data. I believe, that these arguments will make the article more valuable. Otherwise, the data based only on the human memory is speculative.
  2. We appreciate your insightful comments and completely agree with your opinion. We have added the relevant text and references to the discussion.
  • Line 263-274: The mechanisms underlying the link between breastfeeding and diabetes are still unclear. Several probable mechanisms reported in the literature may explain the inverse relationship between breastfeeding and the risk of type 2 diabetes. First, breastfeeding is effective for weight loss and preventing obesity by increasing energy consumption [11], which can help lower the risk of diabetes. Additionally, breastfeeding exerts a positive effect on glucose metabolism by lowering fasting blood sugar and improving insulin resistance in both human and animal studies [12,13]. There are also reports showing that postpartum lactation is associated with increased pancreatic beta-cell function in women with gestational diabetes [10]. Breastfeeding has also been linked to improved lipid metabolism [14] and increased maternal adipokine levels, including peptide YY and ghrelin [26], which may reduce the risk of diabetes and metabolic disorders. These previous reports further support our findings.

Again, thank you for giving us the opportunity to strengthen our manuscript based on your valuable comments and queries. We have worked hard to incorporate your feedback, and we hope that these revisions persuade you to accept our submission.

Reviewer 2 Report

Background

The authors do not provide information on the biological link between breastfeeding and diabetes or how the duration of breastfeeding affects the incidence of diabetes in perimenopausal and postmenopausal women. Are there any studies reporting the effect of breastfeeding and its duration on impaired fasting glucose and diabetes in no menopausal women? What are the results?

Material and methods

The authors do not state which variables were considered as confounders in the multivariate regression analysis

Discussion/ conclusions

My main concern is authors did not control for family history of diabetes and gestational diabetes history. This is a limitation that must be addressed.

Author Response

Reviewers' comments:

Reviewer #2

  1. We appreciate the time and effort you have dedicated to providing insightful feedback on ways to strengthen our paper.

Background

The authors do not provide information on the biological link between breastfeeding and diabetes or how the duration of breastfeeding affects the incidence of diabetes in perimenopausal and postmenopausal women. Are there any studies reporting the effect of breastfeeding and its duration on impaired fasting glucose and diabetes in no menopausal women? What are the results?

  1. We appreciate these insightful comments. You raised an important issue, and we fully agree with you. In response to your comments, we have added further information in the introduction regarding the biological link between reduced diabetes and the duration of breastfeeding in postmenopausal women.
  • Line 48-53: It has been suggested that breastfeeding lowers the risk of subsequent maternal diabetes through several potential mechanisms, e.g., increasing maternal energy expenditure, improving insulin sensitivity and glucose metabolism, and affecting lipid metabolism [10-14]. Previous epidemiological studies based on this premise have investigated the association between breastfeeding and diabetes, and the results showed that women who had never breastfed had an increased risk of diabetes [15,16].
  • Line 64-67: To the best of our knowledge, there has been only one study on the association between the duration of breastfeeding and maternal diabetes in postmenopausal women; that study showed that breastfeeding for more than 3 months lowered the risk of diabetes [18].

Materials and methods

The authors do not state which variables were considered as confounders in the multivariate regression analysis

  1. We appreciate these insightful comments. You raised an important issue, and we fully agree with you. In response to your comments, we have inserted a description of the confounders into the Materials and Methods.
  • Line 154-156: We considered 11 confounding variables (age, income, education, alcohol consumption, smoking, BMI, hypertension, total cholesterol, menstruation, oral contraceptive use, and exercise), for which we adjusted in the regression.

Discussion/conclusions

My main concern is authors did not control for family history of diabetes and gestational diabetes history. This is a limitation that must be addressed.

  1. We appreciate these insightful comments. You raised an important issue, and we fully agree with you. In response to your comments, we have described this as a limitation of our study.
  • Line 304-306: Finally, due to the lack of data, we could not check for a family history of diabetes or a history of gestational diabetes, both of which are closely related to the development of diabetes.
  1. Again, thank you for giving us the opportunity to strengthen our manuscript based on your valuable comments and queries. We have worked hard to incorporate your feedback, and we hope that these revisions persuade you to accept our submission.

Round 2

Reviewer 2 Report

Authors responded correctly to the suggestions made.